# Why Do Women with Eating Disorders Decline Treatment? A Qualitative Study of Barriers to Specialized Eating Disorder Treatment

**DOI:** 10.3390/nu13114033

**Published:** 2021-11-11

**Authors:** Sofie T. Andersen, Thea Linkhorst, Frederik A. Gildberg, Magnus Sjögren

**Affiliations:** 1Department of Studies and Analysis, National Association against Eating Disorders and Self-Harm, 2500 Valby, Denmark; sa@vioss.dk; 2Forensic Mental Health Research Unit Middelfart, Department of Regional Health Research, Faculty of Health Sciences, University of Southern Denmark, 5500 Middelfart, Denmark; t.linkhorst@gmail.com (T.L.); Frederik.Alkier.Gildberg@rsyd.dk (F.A.G.); 3Psychiatric Department Middelfart, Mental Health Services in the Region of Southern Denmark, 5500 Middelfart, Denmark; 4Psychiatric Center Ballerup, 2750 Ballerup, Denmark; 5Department of Clinical Sciences, Umeå University, 901 87 Umeå, Sweden

**Keywords:** eating disorders, anorexia nervosa, barriers, treatment, adult, qualitative, patient perspective

## Abstract

Despite the fact that eating disorders (EDs) are conditions that are potentially life-threatening, many people decline treatment. The aim of this study was to investigate why women decline specialized ED treatment, including their viewpoints on treatment services. Eighteen semi-structured qualitative interviews were conducted with women who had declined inpatient or outpatient specialized ED treatment. A thematic analysis revealed five main themes: (1) Disagreement on treatment needs, (2) rigid standard procedures, (3) failure to listen, (4) deprivation of identity, and (5) mistrust and fear. The women had declined ED treatment because they believed that treatment was only focused on nutritional rehabilitation and that it failed to address their self-identified needs. From their perspectives treatment was characterized by rigid standard procedures that could not be adapted to their individual situations and preferences. They felt that the therapists failed to listen to them, and they felt deprived of identity and reduced to an ED instead of a real person. This investigation is one of the first of its kind to provide clues as to how treatment could be moderated to better meet the needs of women who decline specialized ED treatment.

## 1. Introduction

Eating disorders (EDs) are severe and potentially lethal conditions that have somatic, psychological, and social implications for the afflicted [1,2,3]. Despite these devastating consequences, many individuals with EDs fail to engage with treatment or even decline treatment, should they be referred to a specialist unit [4,5,6,7]. A British study found that only half of those patients who are referred to a specialist eating disorder service entered treatment [6]. Patients who decline treatment put themselves in a potentially self-destructive pattern with sometimes life-threatening situations if they go untreated for extended periods of time. In addition, several studies have shown that early intervention improves treatment outcomes [8,9] and, thereby, one of the most critical decisions that many individuals with EDs have to consider is whether or not to accept treatment.

A few qualitative studies have investigated barriers to initiating ED treatment with a focus on their first encounter with health care services [5,10]. In one of these studies, Leavey et al., (2011) found that some individuals refused to enter treatment because they were ambivalent about behavioral change, and because the ED had become an integral part of their identity for so long that life without such eating patterns seemed unimaginable. In addition, the study found that some patients felt that their expectations of the first appointment were not met, and they were disappointed with the treatment that was on offer [5]. Likewise, a study by Gulliksen et al., (2015) found that, at first appointment with a health care professional, most patients were not motivated to change their ED behavior. Instead, they often desired to reduce somatic symptoms and feel less depressed, which they experienced was not met by the health care providers. This indicates that there may be a discrepancy between the patient perception of their individual problem and associated needs for treatment and service being offered by the health care provider. Previous studies have found that lack of congruence between patients’ and therapists’ expectations of treatment [11] and low treatment credibility [12] are associated with increased risk of dropout from treatment among patients with eating disorders. In addition, a recent study that investigated barriers to initiating inpatient specialized ED treatment found, among other things, that individuals with EDs were quite skeptical about the likelihood that inpatient treatment would be effective in their recovery process [4]. Patients believed that treatment would only help them gain weight, but not result in psychological change [4].

Patients with EDs are known to be ambivalent about their illness, which can at one and the same time be perceived as a burden, but also as their best friend, because it gives them a sense of control and enables emotional avoidance [13,14]. Patients’ ambivalence and resistance to behavioral change have been found to represent significant barriers to help-seeking [15], to recovery motivation [16,17] and constitute one of the reasons for high dropout rates from treatment [18]. Despite a growing body of literature addressing the themes of barriers to help-seeking and dropout from ED treatment, there is only sparse research on the client’s perspective on barriers to initiating ED treatment and their reasons for declining treatment. Patients who decline treatment differ from those who do not seek help at all; the former group has talked to a health care professional about their ED and needs for treatment. Additionally, with regard to the participants in this current study, they also attended the initial assessments at the ED clinic before declining treatment.

In summary, only a few studies have investigated why individuals suffering from EDs decline treatment, and, to the best of our knowledge, no studies have investigated the individuals’ specific barriers related to treatment services. Research into the barriers to initiating treatment from the perspective of the patients declining treatment can contribute with important insights into the perceived treatment needs of this specific group. Such information may help in communication with those who reject a treatment offer, to avoid further barriers. The present study presents results from interviews with women who had rejected a treatment offer. The aim was to gain a better understanding of barriers to specialized ED treatment within this group, and their perspectives on treatment.

## 2. Materials and Methods

### 2.1. Design

This qualitative study was methodologically informed by the basic requirements for empirical research, as described by Herbert Blumer [19], and used semi-structuredinterviews [20,21] to achieve an in-depth insight into and understanding of barriers to specialized ED treatment, from the viewpoint of the individuals declining treatment. The study took place in Denmark and included individuals who were referred by a general practitioner to a specialist unit for EDs and attended the initial assessments but declined to initiate treatment.

### 2.2. Participants

Participants were recruited by way of an advertisement posted at ED treatment units and on the website and social media page of the National Association against Eating Disorders and Self-harm. In addition, patients who declined treatment at a specialist ED unit at Psychiatric Centre Ballerup were contacted by email and invited to participate in the study. Individuals who showed interest first completed a short questionnaire regarding their age, type of ED, location for and type of treatment offered and whether they had attended initial assessments and interviews at the ED clinic. Inclusion criteria were as follows: (1) Referred to outpatient or inpatient treatment at a Danish public ED specialized unit; (2) attended at first assessment at the ED clinic; (3) had declined to initiate treatment; (4) aged 18 or over at the time when treatment was declined and (5) refusal to initiate treatment occurred within the last four years.

Among individuals with interest in the study, 25 met the inclusion criteria. To maximize the potential variation in the informants’ experiences, we selected a heterogeneous sample of participants who differed as much as possible within these criteria (Table 1).

In addition, we used purposeful sampling to select the participants. The aim of purposeful sampling is to select information-rich cases that are especially knowledgeable about or experienced in a phenomenon of interest [22,23]. It should be noted that no individuals suffering from Bulimia Nervosa (BN) showed interest to participate in this study. Data were saturated at the conclusion of 18 interviews; that is, redundant data were generated and varying perspectives had been gathered.

The selected participants were all women, aged between 20 and 46 (Mean age 30.8, SD 8.41). Most were suffering from Anorexia Nervosa (AN; 72%; self-reported diagnosis) and atypical eating disorders (atypical EDs; 28%; self-reported diagnosis). Time since onset of ED was between 3 and 30 years (Mean 15.3 years, SD 7.52; self-reported data) meaning that some of the participants may fit well into a severe and enduring ED category. In total, 72% had been offered treatment at an outpatient ED clinic and 28% had been offered treatment at an inpatient ED clinic. The period between declining treatment and participating in the interview was between 2 and 48 months (mean 16.7, SD 13.39). Of the 18 participants, 15 had previous treatment experiences from the same or other specialized ED units, while three had never been in treatment. At the time of declining treatment, the 18 participants were not ill enough to be involuntarily admitted to treatment against their will. After declining treatment, they chose different ways to deal with their ED; some found help from psychologists, dietitians, or personal trainers, while others found alternative treatment helpful, e.g., hypnosis or body therapy. At the time of the interview, some participants were still searching to find suitable treatment options, while a few told us that they had later chosen to accept a treatment offered at a specialist ED treatment unit since they could not find any other possibilities.

### 2.3. Data Collection

Guided by the following research questions, an interview guide was developed [24] (Table 2): (1) Why do adult women with EDs decline treatment? (2) How should specialized ED treatment be adapted to better meet the perceived treatment needs of individuals who decline treatment? The interview guide was pilot tested and with a therapist working with EDs and a person who declined ED treatment. Both provided feedback which informed further refinement of the questions.

Between August 2019 and January 2020, 18 individual, in-depth interviews were conducted face-to-face, audio-recorded, and transcribed. The duration of the interviews was, on average, 65 min. The interviews were mainly held in the participants’ homes, but a few were conducted at the premises of the National Association against Eating Disorders and Self-harm. The interviews were conducted in Danish by a qualitative researcher (S.T.A.), and a research assistant took notes on facial reactions and body language.

### 2.4. Data Analysis

To answer the research questions, a thematic analysis [25] was carried out, following five steps: Step 1: The interviews were read through and notes were taken, with the intention of identifying overall thematic headlines and structures in the text. Step 2: The analytical questions are then defined. The analytical questions differ from the research questions; the former takes into account the inclusion and exclusion criteria, clarification of concepts, delimitations and other relevant issues depending on the study [25]. The interviews were coded by reading through the interview text, searching for the answer to the analytical questions. All of the text that represented responses to the analytical questions was marked with a subject heading, and an authenticity marking, and was condensed, whereafter a codebook was developed [26]. Step 3: Each subject heading and condensation was subsequently categorized and sorted into themes and a coherent thematic text was developed. Notes from Step 1 were included if relevant to the aim. Step 4: The themes were grouped taxonomically into themes and sub-themes, based on the semantic relation ‘a is a part of b’ [25,27]. Step 5: Finally, themes were then recontextualized by testing them against the original text, in order not to skew interpretations [19,25]. The analysis was supervised by a third author (F.A.G.), conducted by the first (S.T.A.) and second (T.L.) authors and validated by all authors.

### 2.5. Ethical Considerations

Ethical approval was obtained from the Danish Patient Safety Authority (Approval number: 3-3013-2603/1). All participants were fully informed about the aim of the study. They participated voluntarily and gave their written informed consent before participating. The participants were assured confidentiality, anonymity, the right to withdraw consent at any time and that their choice to participate had no bearing on their ability to receive treatment at the specialist ED treatment unit.

## 3. Results

As shown in Figure 1, the analysis resulted in five main themes: ‘Disagreement on treatment needs’, ‘Rigid standard procedures’, ‘Fail to listen’, ‘Deprivation of identity’ and ‘Mistrust and fear’. The themes, their associated sub-themes and their interrelatedness are presented in the section below.

### 3.1. Theme 1: Disagreement on Treatment Needs: “It’s Not about Gaining Weight” (Reported by 16/18 Interviewees)

The participants thought that the treatment they had been offered did not fit with their perceived needs for treatment. The quote below illustrates a common view among the participants, that treatment primarily focused on ED symptoms and weight gain, and therefore might help them to gain weight, but would not be helpful in the long run.

# 14: “I also know that if I enroll [in the treatment program] then I might gain weight and we might even work on a healthy eating pattern. But once that’s over … Well, I might be able to bury my ED in a box for a few years. But then, bang, it hits me again. I really don’t want to spend the rest of my life like that”.

Furthermore, the participants believed that within the treatment program their attention to weight and ED symptoms might even increase. They felt that they needed more therapeutic help and a better understanding of the underlying causes of their ED.

#### Sub-Theme: Lack of Responsibility (Reported by 11/18 Interviewees)

Some of the participants perceived ED treatment as including a high degree of control and a lack of responsibility.

# 1: “It seemed like it was the military, and you are mentally ill, that’s what I felt like. You get stripped from responsibility, simply”.

According to the participants, it would not be helpful to them to lose autonomy and responsibility for their own treatment. Instead, they preferred involvement and shared decision-making regarding treatment, rather than the therapists holding all the power. They suggested that they themselves were to partake in defining goals and success criteria.

### 3.2. Theme 2: Rigid Standard Procedures: “There Was No Room for Me” (Reported by 17/18 Interviewees)

The participants perceived treatment as characterized by rigid standard procedures that could not be adapted to their individual situations, preferences, and goals.

# 12: “I felt that to receive help I had to fit into this non-adjustable box of treatment. I felt that there was no room for me to be honest with myself”.

As illustrated by the above quote, the participants felt that if they accepted the treatment offer, they would have to make an effort to fit into the treatment program rather than vice versa, i.e., that the treatment program would be adapted to them. Some participants expressed specific needs, e.g., in relation to comorbidities or complex life situations, but they felt that no consideration was given to them. The participants wanted alternative treatment options and personalized care adapted to their individual situations and preferences.

#### 3.2.1. Sub-Theme: Losing Valued Daily Activities (Reported by 11/18 Interviewees)

The sub-theme ‘Losing valued daily activities’ centers on a perceived clash between the rigid standard procedures, such as fixed time-points and frequency of treatment, and participants’ wishes to maintain daily activities, such as work, studies or seeing friends.

# 8: “Well, I couldn’t get it [the therapy] to fit with my studies, because we have a lot of classes with compulsory attendance. So, I had to drop out or be on sick leave to enter treatment and you know … I’m not really social and I don’t have a lot of friends. So, entering treatment would mean that I’d not see anyone, apart from those at that place [the ED clinic]”.

According to the participants, it was not an option to negotiate when or how often to go to the clinic, and accepting treatment therefore marked the end of valued daily activities. The participants believed that their daily activities gave them peace of mind from troubling ED thoughts, and that losing these routines might additionally result in them losing their connection to a normal life, thereby exacerbating their ED.

#### 3.2.2. Sub-Theme: Time Pressure (Reported by 9/18 Interviewees)

# 14: “That treatment is limited in time. That’s probably the most important reason for my rejection of treatment. I could not reconcile myself with the fact that it was set how long it would take me to recover”.

As illustrated above, the participants were frustrated about the narrow treatment approach that implied how long it would take to recover. They felt that fulfilling the required weekly weight gains and reaching the treatment goals within the fixed time frame was too stressful and overwhelming.

### 3.3. Theme 3: Fail to Listen: “What about My Perspective?” (Reported by 16/18 Interviewees)

The participants felt that the therapists were not listening to their perspectives, and that their feelings and needs were not regarded as important.

# 4: “They were not listening to what I said. Even when I tried to communicate how I felt, they were ignoring me. A person with an ED can only be in one way and if I said something which didn’t fit with their perspective of a person with an ED, they made it disappear”.

The participants expressed a need for a trusting relationship with a therapist, who would take their perspectives seriously and who would be supportive. They wanted someone in whom they could trust and someone who would show sincere care.

#### Sub-Theme: Poor First Impression (Reported by 12/18 Interviewees)

The theme ‘Fail to listen’ is linked to the sub-theme ‘Poor first impression’, which characterizes the participants’ first encounter with the ED clinic as cold, uncomfortable and disappointing. According to the participants, no-one had asked them why they were there and how they felt, as illustrated in the quote below.

# 6: “Well, I had sought treatment myself, so it was really intimidating that I was not even given the opportunity to tell them [the therapists] why I was there. They were just assuming a lot of things about me, I guess. I remember that I didn’t even get the chance to introduce myself properly, and it felt so intimidating”.

### 3.4. Theme 4: Deprivation of Identity: “Please, See Me as a Whole Person” (Reported by 16/18 Interviewees)

According to the analysis, themes 1–3 lead to theme 4, ‘Deprivation of identity’ (Figure 1). Focus on weight gain and ED symptoms (theme 1) and failing to listen to the patient perspective (theme 3) made participants feel that they were reduced to an ED, not an individual person. In addition, the rigid procedures within treatment (theme 2) made no room for the participants to be themselves.

# 16: “I think the problem is that the focus is only on the ED and they completely forget the individual behind. Everything is about the ED instead of ‘who am I and which values do I have?’ It is really, really difficult to recover from an ED, if it is the only thing that you have and the only thing that others see in you”.

As illustrated in the above quote, the participants felt that the ED symptoms were the only thing that therapists saw and focused on. They felt that within the treatment facilities your weight defines who you are and what you are worth, rather than who you are as a human. They described that their feelings and actions were misunderstood and interpreted as disordered and wrong, e.g., “doing exercise” or “being picky”. Within the treatment facilities, the participants felt reduced to a disease and a mentally ill person; instead, they wanted to be seen as an individual and treated as a whole person.

#### 3.4.1. Sub-Theme: Feeling Different (Reported by 15/18 Interviewees)

Theme 4 connects to the sub-theme ‘Feeling different’, which was characterized by participants feeling that the treatment approach did not include them.

# 9: “I used to believe that psychiatry had room for everyone, and that even if you had a strange and crooked mindset you would still belong there. Then I found out that I did not belong there either. That made me feel lonely because maybe I was the only one in the whole world who felt the way I felt. And it was obviously not okay to feel that way. They told me that ‘we have room for everyone here’—well, not me”.

The participants perceived that treatment was only set up for the “normal” ED patient, not including, e.g., patients who were older than the average, were suffering from a comorbid diagnosis, or had abnormal symptoms not listed in the diagnostic manuals. The participants felt that it was not okay to feel the way they felt, and they felt neglected.

#### 3.4.2. Sub-Theme: Degrading Treatment (Reported by 9/18 Interviewees)

Closely linked to theme 4 was the sub-theme ‘Degrading treatment’, characterized by participants feeling that within the treatment facilities they were being treated disrespectfully.

# 13: “You know, it’s just degrading … It’s like they tighten wires in you, so I become their puppet, right? That’s what I felt like. That they were starting to put something around my arms so that they could control me”.

Some of the participants believed that they had good insight into their own disease and individual challenges, but experienced that decisions were made without them being involved. They felt left without a voice in their own matters and that they had lost their rights as individuals. If they were not doing as expected, e.g., did not comply with the weight gain requirements, they were punished and threatened with being discharged from the treatment. They felt that they were constantly challenged on their personal boundaries.

### 3.5. Theme 5: Mistrust and Fear: “I Didn’t Feel Safe” (Reported by 15/18 Interviewees)

According to the analysis, themes 1–3 lead to theme 5, ‘Mistrust and fear’ (Figure 1). The treatment program did not fit with the participant’s perceived treatment needs (theme 1) and could not be adapted to their individual situations (theme 2), which made them lose confidence in treatment. In addition, when therapists did not listen to their perspectives (theme 3) they lost confidence in the therapists.

# 6: “Actually, I don’t think I trusted them. I didn’t trust that they were doing it good enough or the right thing. They perceived EDs from a very stereotypical point of view and what if that wasn’t me? […] It was probably the right treatment for some patients, but I didn’t trust that they were keeping an eye on me to make sure that it was also the right thing for me”.

Theme 5 is characterized by participants not trusting that the therapists were trying to help them, and that they did not feel safe at the treatment facility. They were afraid to be honest with the therapists, in the belief that it might have negative consequences for them. Some were afraid of being subjected to coercive actions and being placed in a closed ward, not knowing when they would get out. From their point of view, treatment was uncontrollable and unpredictable which made them feel insecure at the treatment facility.

#### Sub-Theme: Fear of Abandonment (Reported by 10/18 Interviewees)

The sub-theme ‘Fear of abandonment’ was characterized by participants who were afraid of being let down by treatment services if they initiated treatment.

# 11: “It is really stressful to know that, within the timeframe, you have to have recovered. And then you have to handle it on your own. What if I’m not? What if I’m not ready to handle it on my own? I would probably start self-harming and losing weight again”.

The participants had doubts about whether the therapists would be there to help them if their ED got worse or if treatment failed. Some had previously experienced that therapists changed agreements at the very last minute or failed to comply with them, which made them feel cheated on. They had fears of abandonment and for some participants, it felt easier to decline help and hold on to their ED.

## 4. Discussion

This qualitative study explored why women with EDs decline specialized treatment and their views on how treatment services could be moderated to better fit their needs.

Previous studies have found that motivating adults suffering from EDs to engage in treatment is complicated, as the desire to recover might co-exist with significant ambivalence toward behavioral change and a significant fear of losing control [10,17]. It is reasonable to assume that many patients who decline to take up specialist ED treatment are not ready or willing to change their behavior. However, the five barriers to initiating treatment presented in the current study are related to treatment services and interactions with therapists, and thus, this study offers another perspective on barriers to initiating treatment.

Our participants perceived the focus on ED symptoms and nutritional recovery as a barrier to initiating treatment. They would prefer treatment that focused on emotional and psychological wellbeing, instead of on the visible and physical signs of the ED. Research into the prognosis of ED, especially AN, has stressed the importance of, at least at the beginning of the treatment, addressing underweight [28], focusing on changing dysfunctional eating behaviors, and successively increasing attention to psychological matters [29]. This study found that weight and eating behaviors may not be the aspects of treatment that individuals who decline treatment prefer. Even though our study selected individuals who rejected treatment, which may reflect the rather negative views they presented on treatment, the findings somewhat echo what other studies have stressed. For example, individuals with EDs prefer treatment to be focused on improving quality of life [30,31] and on emotional and psychological factors, rather than on weight and food intake [32,33,34,35]. In addition, the focus on nutritional rehabilitation might even trigger restricted eating and encourage more ED behaviors [32]. ED patients suffer from impaired quality of life [36], and given that increased body mass index (BMI) may not always reflect an improvement in quality of life [36], greater emphasis on emotional and psychological wellbeing alongside weight gain may contribute to better satisfaction with treatment or maybe even better treatment outcomes.

Another finding from the current study presents an opposition between control and responsibility. Most of our participants preferred involvement and shared decision-making regarding treatment options, rather than the therapists holding all the power in relation to these decisions. Our study does not allow for an inference to actual treatment situations, and we only know the patients’ perspectives of the situations. However, the findings of this study do stress the importance of ensuring shared decision-making within treatment to enable engagement. Other studies have found that, while some patients felt vulnerable and anxious when handing over personal control in treatment, others felt relief, because a highly structured treatment environment gave them a sense of order and certainty [35,37,38]. In addition, these highly structured environments may play a critical role in restoring patients to health and ensuring patient safety [28]. On the other hand, studies have found that recovery motivation in EDs may be improved by helping clients maintain a sense of autonomy over their treatment and by using an empowering approach [16,39,40]. In practical terms, when therapists show respect for the patient as the leading actor in their own lives, therapists can help patients to build motivation for recovery and more readily accept treatment [16,39]. In addition, based on a review of the recent literature on ED treatment outcomes, a study concluded that treatment is most effective when it is collaborative and relies on personal patient autonomy [41]. A treatment approach based on the principles of empowerment and respect for the individual is called the recovery-oriented approach, which has spread widely within mental health services in many countries, including Denmark [42]. Some studies have found that active involvement in decision-making and clinical orientation towards empowerment is associated with higher satisfaction with care among clients [43] and that recovery-oriented treatments are associated with better outcomes in patients with mental illnesses [44]. However, so far, the recovery-oriented approach is not common in ED treatments.

Another major theme was that treatment was perceived as rigid and that the participants experienced standard treatment procedures negatively, because they were considered not to address the person’s self-identified needs. The results revealed that participants wanted tailored interventions and personalized care adapted to their individual preferences. These findings resonate with and reinforce studies that have examined ED patients’ views in relation to eating disorder treatment [31,34,38,45]. Those studies indicate that, even though there is little consensus amongst patients regarding the type of treatment that would be optimal, they prefer when treatment is personalized to their individual needs and preferences.

The results of this study have the following implications for health care professionals working with patients suffering from EDs who reject a treatment offer. The results imply that individuals refusing treatment may prefer:
(1)A focus on emotional and psychological wellbeing more than on weight gain and ED symptoms.(2)To become active agents engaged in their own treatment and to receive support that promotes their personal responsibility and agency.(3)Acknowledgement of individual differences and tailored interventions to patient’s self-identified needs.(4)A therapist who listens to the patient’s perspective.(5)A therapist who sees the ‘whole person’ instead of only focusing on the ED.

The above implications do not preclude that current treatment already includes the proposed elements. Rather, it emphasizes what some individuals with EDs prefer. Overall, the results should as well be considered in light of the below-described limitations of this study.

### 4.1. Limitations

There are some limitations to this study. First, the inclusion criteria find a selection of individuals, namely those who rejected a treatment offer. We decided to use a purposeful sampling strategy [22,23] to select cases for the interview, which meant that we selected those cases who were especially experienced in the phenomenon, instead of trying to achieve generalizability. Spradley (1979) noted the importance of availability and willingness to participate, and the ability to communicate experiences and opinions in an articulate, expressive, and reflective manner, to allow the scientist to achieve a depth of insight into the research questions rather than generalizability of findings. Therefore, the findings of this study must be interpreted with caution and they cannot be generalized to all ED patients. Second, the study is qualitative and retrospective, which means that data are subject to recall bias that might have influenced the opinions of the participants. Third, given that data are a product of the interpersonal engagement between interviewer and participants, and that analysis rests on interpretation, findings are vulnerable to the subjectivity critique applicable to all qualitative research. It should be noted that findings represent an empirically tested interpretation of a real-world phenomenon and should not be considered as objective registrations [19,46]. Fourth, the current study did not have access to health care records and the participants’ diagnoses were thereby self-reported. Fifth, qualitative studies, in general, are limited in transferability of findings to other countries, and findings like the ones presented in the current study must be treated with caution, as this study was conducted at only specialized mental health departments in Denmark. National and international differences in treatment and nursing culture are not accounted for in this study, and that could challenge transferability [47].

### 4.2. Future Research

Because this is one of the first studies of its kind, there is a need to replicate the findings. Future studies would benefit from including more diverse groups of participants, to explore their views, especially men and individuals with BN, but could also be widened to include individuals who have not yet decided whether to decline or accept treatment. In addition, it would be relevant to investigate whether the reasons for declining treatment differ according to the type of ED, disease severity, duration of the ED or different types of treatment settings offered. The perspectives of family members to individuals with EDs on ED health care services would also be a relevant perspective to investigate. Quantitative survey studies should investigate whether the themes that emerged from this study can be generalized to a larger group of ED patients. Future research into ED treatments may benefit from taking the voices and perspectives of those individuals who decline treatment into account in order to develop treatments which better meet the needs of this group of patients.

## 5. Conclusions

To our knowledge, this study is one of the first to investigate why women suffering from EDs decline specialized treatment. The study reveals barriers that prevent adult women from accepting a treatment offer as well as their perspectives on how treatment could be moderated to better meet their needs. The women in this study believed that treatment was only focused on nutritional rehabilitation and that it failed to address their self-identified needs. They believed they needed more psychological and therapeutic help together with more involvement and shared decision-making regarding treatment. Common amongst the interviewees was an impression of the treatment as rigid and that it could not be adapted to their individual situations and preferences. They wanted alternative treatment options and personalized care adapted to their individual situations and preferences. From their point of view, the therapists failed to listen to their perspectives, and they expressed a need for a trusting relationship with the therapists, who would recognize their views and be supportive. Furthermore, in the treatment, they felt deprived of identity and reduced to an ED instead of a real person, and they had mistrust and fears about treatment.

Although the perspectives of the participants in this study may be biased by their illness and ambivalence, the findings provide important insights into the self-identified treatment needs of these patients, whose voices are currently not represented within research. Appreciating their opinions may help to develop treatments which better meet the needs of this group of patients.

## Figures and Tables

**Figure 1 nutrients-13-04033-f001:**
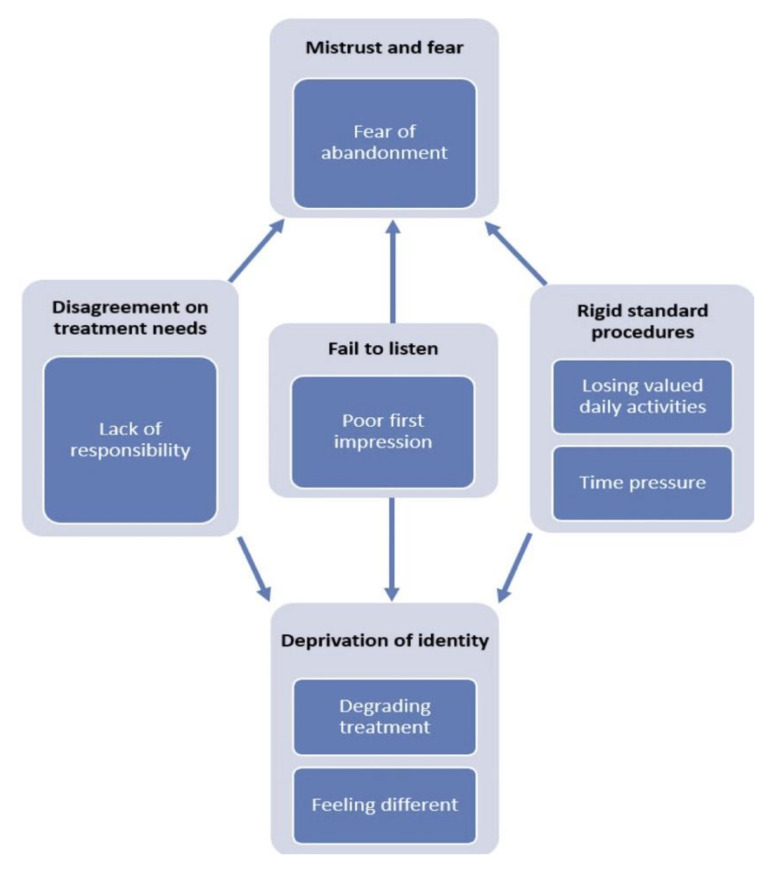
Thematic map of main findings.

**Table 1 nutrients-13-04033-t001:** Participant characteristics (self-reported data).

Variable	Category	Mean/n	(SD) or Percent
Age (years)		30.8	(8.41)
Time since onset of ED (years)		15.3	(7.52)
Time since declining treatment (months)		16.7	(13.39)
Treatment offered	Outpatient	13	72%
	Inpatient	5	28%
Type of ED	AN	13	72%
	Atypical ED	5	28%
Past treatment experiences	Yes	15	83%
	No	3	17%
Place of treatment	Capital Region of Denmark	12	67%
	Region of Nothern Denmark	2	11%
	Region Zealand	1	6%
	Region of Southern Denmark	1	6%
	Central Region of Denmark	2	11%

**Table 2 nutrients-13-04033-t002:** Interview guide.

Question Logic	Question(s)
Current life situation and eating disorder development	Can you tell me a little about yourself? When did you develop an eating disorder?
Perception of illness and motivation for treatment	When did you first seek treatment? Why?
Expectations of treatment	What thoughts and expectations did you have about treatment at this time?
Previous treatment experiences	Have you ever been in treatment at a specialized ED unit before? If yes: How would you describe this experience?
First encounter with specialist ED unit	Now think about the specialist ED unit from where you declined the treatment offered. What happened at the first encounter with the specialist ED unit? How would you describe this experience?
Declining treatment	Why did you choose to decline the treatment you were offered? What considerations did you made?
Suggestions for improvement	From your perspective, how should the specialist ED unit improve to better meet your treatment needs?
Other treatment	What did you do after you declined the treatment offer? Did you get treatment or help from other places, or did you handle the eating disorder on your own?
Situation today	How are you today? What role do your ED have today?

## Data Availability

The data presented in this study are not publicly available, in accordance with the type of consent obtained about the use of confidential data.

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
