# Peer review of "Why Do Women with Eating Disorders Decline Treatment? A Qualitative Study of Barriers to Specialized Eating Disorder Treatment"

_nutrients, 2021, doi:10.3390/nu13114033_

Round 1

Reviewer 1 Report

This is an important topic. Understanding the reasons for failure to engage in treatment is crucial, given the serious nature of eating disorders, the importance of prompt engagement and the poor prognosis without treatment. The abstract is clear and succinct.

Intro

The background literature relevant to this study is reviewed carefully. The aims of the study are clearly stated. A qualitative methodology is appropriate to the aims.  

Research design

Design of the study is appropriate to the aims of the study.  

Methodology

Participants: 18 participants were selected in a process of purposive sampling of 25 possible participants.  The authors could state why they selected the number of 18. Was this due to data saturation with no new themes emerging? In the description of participants, it would be helpful to state whether the participants had received treatment from other centres/voluntary groups/GPs since declining treatment at the specialist unit. They could also mention that this was (presumably) a group that was not ill enough to be compelled to have treatment against their will (using mental health law).

Data collection: it would be helpful to provide more details of the interview guide – what were the topics/questions asked? The relationship between the interviewer and participants needs some discussion. Was there any potential influence on the participants in relation to the characteristics of the interviewer? Was confidentiality assured for the participants?

Data analysis: Description of the process of thematic analysis could be a little clearer for the reader. The authors should clarify the analytical questions and thematic headlines. The process of validation by all authors needs clarification – how was this done?  Did the researchers examine their own role, potential bias and influence during analysis and selection of the data for presentation in the article? This is not clear.

Ethical permission has been granted for the study.

Results

The five main themes and sub themes are summarised clearly in the thematic map. Each theme is illustrated with a relevant quote which highlights the theme.  

Discussion

The authors could discuss and reference research that suggests that weight gain and physical recovery is an essential aspect of eating disorder treatment. There could be more discussion on the tensions between focus on physical and mental health. They could also mention research that suggests that some patients with ED want to be compelled to have treatment. E.g. Research by Tan et al.

Limitations are clearly described. There could be some discussion on why certain individuals chose to take part and some didn’t and how this might have influenced the results.

With respect to further research, it would be worth mentioning the voice of family members in providing another perspective.

There could be more discussion on the clinical implications of the findings.

Conclusions These are clearly summarised.

Style (clarity/typos)

Generally clear style of writing.

Typo. Theme 3.4 line 11 “therapists so and focused on “ .   Should this be “therapists saw and focused on”?

Author Response

Dear reviewer 1, thank you for your comments and suggestions to improve our manuscript. We have tried to answer your questions as clear as possible and we have revised the manuscript according to most of your suggestions. 

Reviewer 2 Report

Overall, I think it is a curious and entertaining work to read, being a qualitative interview study and with a low sample size. Eating disorders are very prevalent in industrialized countries and devastating not only for the health of the sufferer but also for his/her social-context. It is always helpful to get closer to the reasons for those who do not want to follow a psychological treatment. However, I would like to discuss some points with the authors to improve the work. 

Although the introduction is well written, it could be reorganized and summarized, at times it seems that the ideas are placed as a copy and paste.

Regarding material and methods, the selected cohort is representative of the ED population? could you give examples of atypical EDs? do you have any reference that the interview design followed some structured and validated steps?

In the results, the description of each theme and sub-themes is very interesting, but how were the themes and sub-themes extracted to generate Figure 1? What are the criteria to consider them as themes?

In the discussion, the role of the limitations of the study is welcome. However, I miss that the authors discuss new proposals for intervention in these cases where patients reject the psychological treatment and how it could be addressed. On the other hand, it would be necessary to emphasize the importance of this study versus quantitative studies in EDs, to learn more about the intervention.

Other comments:

- Line 350, "BMI" is not previusly defined.

Author Response

Dear reviewer 2, thank you for your comments and suggestions to improve our manuscript. We have tried to answer your questions as clear as possible and we have revised the manuscript according to some of your suggestions. 

Reviewer 3 Report

The present work qualitatively evaluates why people who suffer from an eating disorder often do not accept the proposed treatment. The study results evidenced the presence of five main themes: disagreement on treatment needs, rigid standard procedures, failure to listen, deprivation of identity, and mistrust and fear.
Overall, the aim of the study is important and interesting. Nevertheless, important limitations are present, which strongly prevent the generalizability of the results.

Major points of concern:
1- The experimental sample is very small and heterogeneous and fails to interpret the differences in the approach to the treatment of patients with different eating disorders. Given that observation, I do not believe that a transdiagnostic approach is correct for a study like this. The clinical experience teaches that patients with different eating disorders have a very different approach to the treatment and different drop-out rates as well. 

2- I do not understand why the authors refer to eating disorders in general throughout the paper since only patients with AN were included in the final sample. Were patients with binge eating disorder considered in the enrollment phase? 

3- I think that including patients with self-reported diagnoses is a major issue of this research, especially considering that we are talking about a group of patients who are not engaged in a therapeutic program. Also, in my clinical experience and opinion, the diagnosis of an atypical eating disorder is challenging, even for an expert consultant. 

4- The included patients have very long disorder durations, making (some of) them de facto patients with Severe and Enduring Anorexia Nervosa. Although this may be an element of interest, it should be specified and should at least be included in the main research objectives. 

5- Much clinical information is missing, i.e., the presence of binge-purging behaviors, current BMI, lowest lifetime BMI, age of onset, years of education, etc. 

6- We know that eating disorders are associated with frequent comorbidities, which can greatly influence the approach to treatment. Although it is statistically predictable that some of the included patients have comorbid psychiatric disorders, these are neither investigated nor specified throughout the paper. 

7- A lot of information about the treatment that was proposed to the patients is lacking. The authors refer to a rigid setting but do not specify what kind of setting it is. Is it a cognitive-behavioral therapy? Nutritional rehabilitation? Multidisciplinary treatment? 

Author Response

Dear reviewer 3, thank you for your comments and suggestions to improve our manuscript. We have tried to answer your points and questions as clear as possible and we have revised the manuscript according to some of your suggestions. 

Round 2

Reviewer 3 Report

I thank the authors for their responses to my comments.

Thanks to their answers, I have better understood some aspects of the study that seemed unclear to me.

I have no methodological criticism different from the previous review. Thus, I still think that the paucity of clinical data and the way sample characteristics are reported make the study not very interesting for the clinical and scientific audience. 

Given these considerations, I leave the decision on the acceptance of the work to the editor.